# Osteosarcoma Cells and Undifferentiated Human Mesenchymal Stromal Cells Are More Susceptible to Ferroptosis than Differentiated Human Mesenchymal Stromal Cells

**DOI:** 10.3390/antiox14020189

**Published:** 2025-02-06

**Authors:** Yuliya D. Smirnova, Dominik Hanetseder, Lukas Derigo, Andreas Sebastian Gasser, Annette Vaglio-Garro, Simon Sperger, Regina Brunauer, Olga S. Korneeva, Johanna Catharina Duvigneau, Darja Marolt Presen, Andrey V. Kozlov

**Affiliations:** 1Ludwig Boltzmann Institute for Traumatology, The Research Center in Cooperation with AUVA, 1200 Vienna, Austria; dyd16@mail.ru (Y.D.S.); dominik.hanetseder@trauma.lbg.ac.at (D.H.); office@lukasderigo.at (L.D.); gasser_andreas@gmx.at (A.S.G.); annette.vaglio@trauma.lbg.ac.at (A.V.-G.); simon.sperger@trauma.lbg.ac.at (S.S.); regina.brunauer@trauma.lbg.ac.at (R.B.); darjamarolt@gmail.com (D.M.P.); 2Austrian Cluster for Tissue Regeneration, 1200 Vienna, Austria; 3Laboratory of Metagenomics and Food Biotechnology, Voronezh State University of Engineering Technologies, 394036 Voronezh, Russia; korneeva-olgas@yandex.ru; 4Institute for Medical Biochemistry, University of Veterinary Medicine Vienna, 1210 Vienna, Austria; catharina.duvigneau@vetmeduni.ac.at

**Keywords:** ferroptosis, osteosarcoma, Erastin, RSL3, Ferrostatin-1

## Abstract

Current research suggests that promoting ferroptosis, a non-apoptotic form of cell death, may be an effective therapy for osteosarcoma, while its inhibition could facilitate bone regeneration and prevent osteoporosis. Our objective was to investigate whether the susceptibility to and regulation of ferroptosis differ between undifferentiated (UBC) and differentiated (DBC) human bone marrow stromal cells, as well as human osteosarcoma cells (MG63). Ferroptosis was induced by either inhibiting glutathione peroxidase 4 (GPX4) using RSL3 or blocking all glutathione-dependent enzymes through inhibition of the glutamate/cysteine antiporter with Erastin. Lipid peroxidation was assessed using the fluorescent probe BODIPY™581/591C11, while Ferrostatin-1 was used to inhibit ferroptosis. We demonstrate that neither Erastin nor RSL3 induces ferroptosis in DBC. However, both RSL3 and Erastin induce ferroptosis in UBC, while Erastin predominantly induces ferroptosis in MG63 cells. Our data suggest that ferroptosis induction in undifferentiated hBMSCs is primarily regulated by GPX4, whereas glutathione S-Transferase P1 (GSTP1) plays a key role in controlling ferroptosis in osteosarcoma cells. In conclusion, targeting the key pathways involved in ferroptosis across different bone cell types may improve the efficacy of cancer treatments while minimizing collateral damage and supporting regenerative processes, with minimal impact on cancer therapy.

## 1. Introduction

Ferroptosis is the most recently discovered form of programmed cell death [1,2]. It is characterized by the formation of specific death messengers during redox reactions between iron-containing compounds and membrane lipids, resulting in increased membrane density and loss of mitochondrial cristae [3]. A growing body of literature suggests that inducing ferroptosis is a promising strategy for cancer therapy [4]. For instance, recent studies suggest that ferroptosis may slow down the growth of osteosarcoma [5]. This assumption is supported by numerous publications in the past year [6,7,8]. Since osteosarcoma is one of the most aggressive malignancies of bone tissue [9], the search for new therapeutic approaches to treat this disease is of great interest. Some drugs, such as sulfalazine [10], tirapazamine [11], bavachin [12], Erastin, and RSL3 can also be used for treatment of rhabdomyosarcoma [13,14], fibrosarcoma [15], and osteosarcoma [16,17]. Conversely, induction of ferroptosis has been shown to be deleterious if induced in normal cells. Numerous studies in recent years have shown that excessive ferroptosis in normal tissues can lead to the development of various diseases, such as cardiovascular diseases [18], Alzheimer’s disease [19], Parkinson’s disease [20], atherosclerosis [21], and others. It has also been implicated in the development of osteoporosis [22,23], although the exact mechanisms remain elusive [24]. It is not yet clear whether the pathways inducing ferroptosis in osteosarcoma and normal cells are the same or not, pointing to the importance of investigating the mechanisms that activate or inhibit ferroptosis in different types of healthy and diseased cells. Some suggest that ferroptosis in osteoblasts, rather than differentiated osteocytes, might be the primary cause of osteoporosis [25]. Additionally, iron overload has been shown to inhibit osteogenesis and poses a high risk for the development of osteoporosis [26], although the underlying mechanisms are still not fully understood.

One of the classical regulators of ferroptosis is the Cystine/Gluthathione (GSH)/Glutathione peroxidase 4 (GPX4) axis, with GPX4 acting as an antioxidant enzyme reducing cytotoxic lipid peroxides to non-toxic lipid alcohols. Increased levels of lipid peroxides generated by lipoxygenase are the major factor contributing to the induction of ferroptosis. This axis is dependent on cystine supply, which is provided by the the Xc system, a cystine/glutamate antiporter that imports cystine into cells to produce cysteine and glutathione. Thus, this antiporter is critical for the GPX4 activity since glutathione is the cofactor of this enzyme [27]. The entire axis can be targeted with Erastin and RSL3, which are able to inhibit the activity of the Xc- or the GPX4 system, respectively, leading to excessive accumulation of lipid peroxides (LPO) on cell membranes and subsequent ferroptotic cell death [28]. It has been previously shown that osteocytes cultured in diabetic periodontitis exhibit suppressed GPX4 expression [29]. In MG63 osteosarcoma cells, decreased GPX4 expression is also associated with the progression of ferroptosis [30].

Induction of ferroptosis can be abolished by Ferrostatin-1, a selective inhibitor of ferroptosis. Ferrostatin-1 is an aromatic amine [31] that specifically binds to lipid-reactive oxygen species associated with ferroptosis [27] and directly inhibits Lipoxygenase (LOX), an enzyme synthetizing lipid peroxides required for induction of ferroptosis [32].

More recently, an alternative to the GPX4 pathway of ferroptosis has been described. This pathway is mediated by glutathione S-Transferase P1 (GSTP1), which is a cytosolic enzyme, which exerts peroxidase activity directly converting lipid hydroperoxides into non-toxic, lipid alcohols in a selenium-independent manner [33]. It has been shown that GSTP1 prevents irradiation-induced death of pancreatic cancer cells by inhibiting ferroptosis [34] and the increased expression of this enzyme is associated with cancer risk [35] and particularly with osteosarcoma. Meta-analysis of data on osteosarcoma revealed that GSTP1 polymorphisms may be an important risk factor for osteosarcoma [36] and that GSTP1 null genotype is associated with the higher risk of osteosarcoma [37]. These data suggest that GSTP1 may be important for regulation of ferroptosis in osteosarcoma and may be in other types of bone cells. The regulation of the cystine/GSH/GPX4/GSTP1 axis instead of iron-containing drugs remains a relevant therapeutic target for the induction of ferroptosis and become a basis for potent therapy. It is known that the efficiency of chemotherapy is associated with the induction of ferroptosis [38] due to the delivery to the target tissue toxic amounts of iron [39], which is accompanied by very heavy side effects on the patient.

The study of targeted activators and inhibitors of ferroptosis still requires more detailed investigation, especially in the context of the effect on healthy cells surrounding the tumor. In this sense, the stratification of the sensitivity of different cell types to ferroptosis using a fluorescent reporter for lipid peroxidation is one of the promising methods in the development of a therapeutic approach [40].

With regard to bone tissue, the induction of ferroptosis in cancerous bone cells and the inhibition of ferroptosis in healthy bone cells may provide therapeutic approaches for bone cancer and osteoporosis, respectively. However, uncertainties persist regarding the sensitivity and potential differences in the pathways inducing ferroptosis in cancerous cells, and undifferentiated and differentiated hBMSCs. These uncertainties could serve as the basis for therapeutic strategies aimed at promoting bone regeneration without enhancing cancer progression or selectively eliminating cancer cells neither compromising bone regeneration nor damaging differentiated hBMSCs.

The aim of this study was to elucidate the pathways of ferroptosis induction in undifferentiated hBMSCs (UBC), hBMSCs differentiated to osteoblasts (DBC), and osteosarcoma cells (MG63 cell line). To address this aim, we investigated the effectiveness of GPX4 and glutamate/cystine antiporter in the induction of ferroptosis in those three types of bone cells.

## 2. Materials and Methods

### 2.1. Cell Culture

We used cell lines and primary cells: differentiated and undifferentiated human bone marrow mesenchymal stromal cells, and osteosarcoma cell line MG63. MG-63 cells were bought from ECACC (Salisbury, UK) and cultured in DMEM containing 10% fetal calf serum and 1% L-glutamine. Bone marrow mononuclear cells of a 20-year-old female donor were purchased from Lonza (Basel, Switzerland), and stromal cells were isolated by plastic adherence following the manufacturer’s instructions. hBMSCs were cultured in MSC growth medium consisting of DMEM-HG supplemented with 10% fetal bovine serum (FBS), 1% 100 U/mL Penicillin + 100 μg/mL Streptomycin (pen-strep), 2 mM L-glutamine, and 1 ng/mL basic fibroblast growth factor (bFGF). Cultures were incubated at 37 °C and 5% CO_2_ with media changes twice per week. For DBC, hBMSCs were cultured prior the experiment in osteogenic medium for 14 days consisting of DMEM-HG supplemented with 10% FBS, 2 mM L-glutamine and 1% pen-strep, 10 nM dexamethasone, 50 µM ascorbic acid-2-phosphate, and 10 mM ß-glycerophosphate [41]. Passage 4 of hBMSCs were used in all experiments. All cells were seeded into clear, flat bottom 12-well plates at a density of 70,000 cells/cm^2^ 24 h prior to the experiment.

### 2.2. Ferroptosis Induction and Rescue

The cells were treated with either Erastin (0.75, 1, and 2 μM) or RSL3 (0.75, 2.5, and 5 μM) and incubated for 24 h at 37 °C. To dissect cells dying through ferroptosis, Ferrostatin-1 (10 μM) was added to the cells treated with Erastin or RSL3. As a result, the following experimental groups were examined to understand the action of Erastin: 1—control; 2—Erastin 2 μM; 3—Erastin 1 μM; 4—Erastin 0.75 μM; 5—Erastin 2 μM + Ferrostatin 10 μM; 6—Erastin 1 μM + Ferrostatin 10 μM; 7—Erastin 0.75 μM + Ferrostatin 10 μM; and 8—Ferrostatin 10 μM. Similarly, we designed experimental groups with RSL3: 1—control; 2—RSL3 5 μM; 3—RSL3 2.5 μM; 4—RSL3 1 μM; 5—RSL3 5 μM + Ferrostatin 10 μM; 6—RSL3 2.5 μM + Ferrostatin 10 μM; 7—RSL3 1 μM + Ferrostatin 10 μM; and 8—Ferrostatin 10 μM. In addition, we used two inhibitors of GSTP1, auranofin (5 µM) and Piperlongumine (15 µM), as well as an inhibitor of RSL3, cisplatin (10 and 20 µM). The sources of reagents were RSL3 (S8155-5 mg, Eubio, Vienna, Austria); Ferrostatin-1 (S7243-5 mg, Eubio, Austria); Erastin (S7242-5 mg, Eubio, Austria); Auranofin (A6733-10 mg, Sigma, Darmstadt, Germany); Piperlongumine (528124-25 mg, Calbiochem, Darmstadt, Germany); and Cisplatin (232120—50 mg, Calbiochem, Germany).

### 2.3. LDH Analysis

Lactate dehydrogenase (LDH) is an enzyme released into the extracellular space upon disintegration of the cellular membrane and can be utilized as a marker for cytotoxicity, cell injury, and cell death. Its activity can be detected via conversion of lactate to pyruvate, with concomitant reduction of NAD+. If the reduction of NAD+ is coupled to the reduction of tetrazolium salt (INT) with phenazine methosulfate (PMS) serving as an intermediate electron carrier, only catalytic quantities of the enzyme are required, as NADH can be reoxidized by PMS back to NAD+. Because NADH converts the yellow INT to red formazan, the color change is proportional to the amount of LDH released. By measuring the increase in absorbance at 492 nm, it is possible to measure increases in the number of damaged and dead cells compared to controls.

LDH activity measurements were performed as previously described by Weidinger et al. [42]. Briefly, LDH release was measured by mixing thirty microliters of cell culture supernatant with one hundred microliters of LDH assay reagent containing 110 mM lactic acid, 1350 mM nicotinamide adenine dinucleotide (NAD), 290 mM N-methylphenazonium methyl sulfate (PMS), 685 mM 2-(4-iodophenyl)-3-(4-nitrophenyl)-5-phenyl-2H-tetrazolium chloride (INT), and 200 mM Tris (pH 8.2). Absorbance changes were read kinetically at 492 nm for 60 min using a Polarstar Omega plate reader (BMG Labtech, Ortenberg, Germany). LDH activity values were determined as the maximum rate of NADH formation, determined by the changes of absorbance at 492 nm (A492/dt (min)). Measurements were carried out in triplicates. Data were exported to Microsoft Excel (Microsoft Corp., Redmond, WA, USA) for processing and analyzed in Prism 9.2 (Graphpad, San Diego, CA, USA). The release of LDH was very sensitive to small changes in the concentrations of ferroptosis inducers. To account for these variations, we additionally calculated for each inducer the area under the dose curve, an integrative parameter, which is less sensitive to variations with single concentrations (Figure 1A).

### 2.4. L-ROS-Staining

Lipid peroxidation is one of the main factors activating ferroptosis. To assess the level of LPO, we used a sensitive fluorescent probe BODIPY™ 581/591 C11 (Invitrogen, Waltham, MA, USA), which selectively sensitizes LPO in cell membranes (Figure 1B). When BODIPY is oxidized in living cells, the fluorescent emission peak shifts from red (~590 nm) to green (~510 nm), which makes it possible to analyze lipid peroxidation using fluorescence microscopy (See Figure 1B). A stock solution of C11-BODIPY581/591 was prepared in DMSO at the concentration of 1 mM. Once prepared, the stock solution was aliquoted to normal use volumes and stored at −20 °C, protected from light. Cells were stained with 5 µM concentration of BODIPY for 30 min at 37 °C, then the cell monolayer was washed with 1 mL PBS 1X, then five-hundred microliters of HBSS were added and the cells were analyzed immediately. Imaging was performed by a Zeiss LSM510 Meta laser scanning confocal microscope (Zeiss, Oberkochen, Germany), with ex/em of 581/591 nm to detect non-oxidized state, and ex/em of 488/510 nm for oxidized state with a 10× lens. Images (three images per well) were acquired in ZEN 2009 (version 6.0.303, Zeiss, Germany) and processed with ImageJ (version 1.53, U.S. National Institutes of Health, Bethesda, MD, USA). The ratio 510/(510 + 591) fluorescence was taken as a measure for LPO.

### 2.5. Cell Morphology

Before and after incubation of cells with Erastin, RSL3, and ferrostatin (please see different treatment groups), cell morphology was assessed using a LSM510 Meta laser scanning confocal microscope (Zeiss, Jena, Germany) with a 10× lens.

### 2.6. Statistical Analysis

All measurements were performed in quadruplicates. Data are reported as mean ± SEM. Statistical analysis was performed using GraphPad Prism version 9.2, using one way ANOVA for not matched samples and RM-ANOVA for samples matched by plate containing treated samples and corresponding controls. ANOVA was followed by a post hoc Holm–Sidak’s, a power multiple comparisons test. Other details are indicated in figure legends.

## 3. Results

In all three cell types, ferroptosis was induced by either Erastin, an inhibitor of the glutamate/cysteine antiporter, or RSL3, a specific inhibitor of GPX4. Cells were analyzed 24 h after treatment. The induction of ferroptosis was determined by monitoring the release of LDH and lipid peroxidation (LPO) using the fluorescent reporter, BODIPY (Invitrogen, Waltham, MA, USA).

Initially, we assessed whether there were differences in the release of LDH and LPO in untreated cells 24 h after medium change. No significant changes in LDH release were observed between UBC, DBC, and MG63 (Figure 2A). However, levels of LPO were notably lower in DBC compared to UBC and MG63 (Figure 2B).

Subsequently, all three cell types were treated with either Erastin or RSL3, and LDH/LPO levels were determined after 24 h. In UBC, Erastin treatment resulted in characteristic morphological changes, with the shape of cells becoming round, a typical sign of cell death, accompanied by a reduced number of cells (Figure 3A left). In contrast, combined treatment with Ferrostatin-1 and Erastin resulted in the maintenance of normal cell morphology, suggesting that Ferrostatin-1 inhibited the action of Erastin (Figure 3A right). Treatment with Erastin resulted in an approx. six-fold increase in LDH release vs. DMSO control group and was comparable between the three tested concentrations (Figure 3B). Addition of Ferrostatin-1 reversed the changes induced by Erastin, with LDH release levels comparable to those of the control. The elevated levels of LDH release upon Erastin treatment was accompanied by an increase in LPO levels (Figure 3C,D), whereas addition of Ferrostatin-1 abolished this increase.

Treatment of UBC with RSL3 resulted in responses similar to those observed with Erastin (Figure 4). Morphological changes characteristic of cell death were not observed in the control group, but appeared upon treatment with all three concentrations of RSL3 (Figure 4A), accompanied by an elevated release of LDH into the medium (Figure 4B). These changes were reversed by the addition of Ferrostatin-1 (Figure 4A,B). Additionally, LPO levels were elevated and returned to normal values upon treatment with Ferrostatin-1 (Figure 4C,D). Thus, undifferentiated BMSC respond similarly to Erastin and RSL3 in terms of ferroptosis induction.

Treatment of DBC with Erastin resulted in a different response compared to UBC (Figure 5). No significant changes in cell morphology were detected upon Erastin treatment. Also, the LDH assay did not reveal any changes in response to Erastin. In addition, in differentiated hBMSCs, no effect of Erastin on LPO was detected in differentiated hBMSCs (Figure 5C,D), supporting the assumption that they do not undergo ferroptosis. The oxidation levels of lipids were not changed. Together, these data suggest that differentiated hBMSCs do not respond to Erastin with ferroptosis.

Treatment with RSL3 induced minor morphological changes in DBC, with some cells displaying a round shape and others slightly smaller size than control cells (Figure 6A). A significant increase in the cell death was observed at two RSL3 concentrations, 1 and 5 μM (Figure 6B). It is also interesting to note that in this case, Ferrostatin-1 was not able to abolish the effect of RSL3, suggesting a different pathway of cell death than ferroptosis, which, however, appeared in a very small extend compared to the induction of ferroptosis in other types of bone cells. Significant changes in LPO levels were observed at the lowest RSL3 concentration (1 μM) (Figure 6C,D). In this case, there was a statistically significant increase in oxidized lipids compared to the control.

MG63 cells exhibited very high sensitivity to Erastin-induced ferroptosis (Figure 7). Similar morphological changes in UBC were observed, with a substantial increase in the LDH release and elevated LPO levels. All concentrations of Erastin induced approx. a ten-fold increase in the rate of cell death, which was accompanied by a strong increase in LPO levels. Treatment with Ferrostatin-1 reversed these changes.

The response to RSL3 was much less pronounced compared to Erastin but we observed a change in the cell shape. As can be seen in Figure 8A, RSL3 led to major visible changes in cell morphology at all three concentrations (Figure 8A), but we did not determine any significant increase at any of the tested RSL3 concentrations (Figure 8B). Significant changes in LPO levels were noted at the highest concentrations of RSL3 (2.5 and 5 μM) (Figure 8C,D). These changes were abolished by Ferrostatin-1 (Figure 8D).

Our results show that small differences in the concentrations of ferroptosis inducers result in quite different responses between experiments. This makes it difficult to compare experimental settings based on single concentrations of ferroptosis inducers. To facilitate reliable comparisons across experiments, we calculated the area under dose-response curve for each experiment, encompassing all concentrations of ferroptosis inducers. This enabled comprehensive analysis and comparison of parameters across different experimental settings.

Figure 9 illustrates the AUDC data on LDH in response to Erastin (Figure 9A–C) and RSL3 (Figure 9D–F) in UBC (Figure 9A,D), DBC (Figure 9B,F), and MG63 (Figure 9C,E). Notably, both Erastin and RSL3 induced similar changes in undifferentiated cells UBC (Figure 9A,D), resulting in approximately a six-fold increase in LDH release, which was mitigated by supplementary treatment with Ferrostatin-1. DBC did not respond to Erastin, whereas RSL3 showed a trend toward increased cell death and lipid peroxidation levels, which was abolished by Ferostatin-1. Thus, the analysis of the AUDC revealed weak sensitivity of DBC to RSL3, but the response was marginal compared to UDC and was not affected by Ferrostatin-1.

An interesting difference in response to ferroptosis induction was observed in MG63 cells. The effect of Erastin was very strong, while response to RSL3 was similar to DBC, although it was sensitive to Ferrostatin-1 (compare Figure 9E and Figure 9F). When we compared the responses of all three cell types to Erastin and RSL3, only in MG63 cells were these responses significantly different between these two inducers across all three types of cells. Thus, we have shown that each of the three types of cells have specific patterns of response to the inducers of ferroptosis.

Figure 10 depicts the effect of Erastin and RSL3 on LPO. In UBC, both inducers resulted in an increase in LPO levels. Although the response to RSL3 was lower, there was no significant difference to Erastin (Figure 10A,D). The increase in LPO was abolished by Ferrostatin. We did not find any significant changes in LPO in DBC, although there was a strong trend in response to RSL3 (Figure 10B,E). MG63 cells exhibited a distinct reaction compared to UBC and DBC, with Erastin inducing significantly higher LPO levels compared to RSL3 (Figure 10C,F).

To better understand the mechanism of ferroptosis induction in MG63 cells we additionally tested one inhibitor of GPX4 (cisplatin) and two inhibitors of GSTP1 (Auranofin, and Piperlongumine) in this cell line. The data are displayed in supplement. We did not observe any induction of cell death by cisplatin (Appendix A), while both GSTP1 inhibitors strongly elevated cell death (Appendix A), which was only partially inhibited by ferrostatin, in contrast to classical ferroptosis inducers, RSL3 and Erastin.

## 4. Discussion

In this study, we investigated the susceptibility to ferroptosis of three types of bone cells: undifferentiated hBMSCs, hBMSCs differentiated to osteoblasts, and osteosarcoma cell lines. Ferroptosis was induced with either Erastin or RSL3, inhibitors of the glutamate/cysteine antiporter and glutathione peroxidase-4, respectively. We assessed lipid peroxidation induction and cell death rates in all three cell types. A combination of RSL3 and Erastin allowed us to dissect the impact of GPX4 and GSTP1 on the regulation of ferroptosis. To determine the impact of LOOH formation on the regulation of ferroptosis, we determined that LPO levels were determined using BODIPY-C11, which is predominantly indicative of LOOH [43], and other LPO products could potentially be detected.

To dissect the impact of ferroptosis on cell death, we utilized Ferrostatin-1, which binds to Lipoxygenase and disrupts its catalytic activity, thus preventing the formation of lipid peroxides (LOOH) [32]. Our consistent observation that Ferrostatin-1 abolished the increase in LPO levels across all experimental conditions suggests that LOX is the primary source of LOOH in bone cells.

Our observations revealed differences in the baseline levels of LPO among untreated cells (incubated for 24 h in standard medium). Specifically, LPO levels were significantly higher in UBC and MG63 cells compared to DBC, with no difference observed between UBC and DBC. This suggests that a lower rate of LOOH formation in DBC can be the reason for their resistance to the induction of ferroptosis. This observation is in line with a study by Chien-Tsun Chen et al. which showed that osteogenic differentiation of hMSCs was accompanied by a dramatic decrease in intracellular ROS and upregulation of antioxidant enzymes [44]. Such changes in the rate of LOOH synthesis are possible due to elevated levels of antioxidants, low LOX activity, or alterations in lipid composition. However, further investigations are necessary to elucidate the reason for these differences.

We also found that the cells exhibited high sensitivity to small differences in the concentrations of ferroptosis inducers. To ensure robust conclusions, we analyzed not only single concentrations of Erastin/RSL3 but also the area under the dose-response curve, as previously described [45]. Using AUDC, we identified distinct responses of all three cell types to ferroptosis inducers, as summarized in Figure 11.

Both Erastin and RSL3 induced ferroptosis in UBC similarly, indicating that GPX4, which is sensitive to both inducers, predominantly regulates ferroptosis induction in UBC (Figure 11). In contrast, DBC derived from UBC exhibited resistance to ferroptosis induction. Erastin failed to increase LPO levels or induce substantially the cell death, while RSL3 showed a marginal, albeit non-significant (*p* = 0.0575), increase in LPO levels accompanied by a slight, non-significant (*p* = 0.0578) increase in cell death. This suggests that ferroptosis in DBC is not fully executed, likely due to low levels of LOOH (Figure 11). The cell death rate in DBC was very low and did not undergo the ferroptotic pathway, since it did not respond to Ferrostatin-1 treatment. We did not identify the mechanism of cell death, which is a limitation of our study.

The induction of ferroptosis in MG63 had specific patterns which were different compared to UBC. In MG63 cells, Erastin induced cell death and LPO at a significantly higher extend compared to UBC and compared to treatment of MG63 with RSL3. This suggests that ferroptosis induction in MG63 cells is predominantly controlled by peroxidase activity of GSTP1 (Figure 11). The effect of GPX4 inhibitor (cisplatin) on osteosarcoma cells is shown in Appendix A. However, in this case, we did not observe any induction of cell death by this substance, further supporting our conclusion that GSTP1 most likely plays an important role in the induction of cell death in osteosarcoma. These suggest that the induction of GPX4-mediated ferroptosis may operate in MG63, but it is only a weak mechanism compared to the induction via GSTP1. This finding is in line with the previous reports suggesting a close association between GSTP1 expression and tumor development. In several tumor tissues, >90% of active GSTs is due to GSTP1 activity, which is highly expressed in various types of cancer [46], including osteosarcoma cells [47], whereas in healthy tissues, among them bone cells, osteocytes [48], GSTP1 expression is usually low [33]. We assume that GSTP1 can protect osteosarcoma cells and increase their resistance to chemotherapy. Data showing the effect of specific GSTP1 inhibitors on osteosarcoma cell death are presented in Appendix A, showing that both inhibitors strongly activate cell death, which is only partially inhibited by ferrostatin, in contrast to classical ferroptosis inducers. We assume that GSTP1 may also induce other cell death mechanisms. It is not likely that data on osteosarcoma can be extended to other cancer cells. For instance, a study by Joo-Won Kim et al. showed that RSL3 was more effective in inducing ferroptosis in lung cancer cells than Erastin and that the sensitivity of these cells to ferroptosis was directly dependent on the level of GPX4 expression [49]. Research examining the effects of Erastin and RSL3 on osteosarcoma cells is still very limited.

The induction of ferroptosis in bone cells can be either beneficial if induced in cancer cells or deleterious if induced in normal cells, as the latter can slow down regeneration and contribute to osteoporosis. Iron is a cofactor for many enzymes involved in collagen synthesis and extracellular matrix formation, both of which are essential for bone regeneration. However, elevated iron levels can suppress osteoblast activity, resulting in weaker bones and reduced regenerative capacity [50]. It has been shown that these negative effects are due to the activation of ferroptosis in bone cells. While it was previously unclear whether this is due to impaired functionality of differentiated osteoblasts or failure of undifferentiated cells to initiate regeneration, our data suggest that ferroptosis induction in undifferentiated bone cells, rather than differentiated osteoblasts, is the key factor involved in compromised tissue regeneration.

In conclusion, our findings demonstrate that differentiated hBMSCs are less susceptible to ferroptosis, whereas undifferentiated hBMSCs and the MG63 cell line are more prone to it. This is in line with several previous studies. Zuli Wang et al. showed that undifferentiated cancer stem cells are more sensitive to ferroptosis than differentiated ones [51]. A study by Géraldine Cuvelier et al. showed that differentiated epithelial cells are resistant to ferroptosis induced by highly peroxidizable conjugated linolenic acids, in contrast to proliferating cancer cells [52]. Furthermore, a study by Xiangze Li et al. showed that BMSCs affected by neuroblastoma were more sensitive to ferroptosis than healthy BMSCs [53]. All this is consistent with our data and suggests that locally targeted induction of ferroptosis in cancer cells can be considered as a basis for a therapeutic strategy. Importantly, the pathways of ferroptosis induction appear to differ between UBC and MG63. This raises the prospect of discriminating between cancerous and healthy bone cells based on ferroptosis induction in cancer and ferroptosis inhibition in stem cells to precisely erase cancer cells or facilitate bone regeneration, respectively.

## Figures and Tables

**Figure 1 antioxidants-14-00189-f001:**
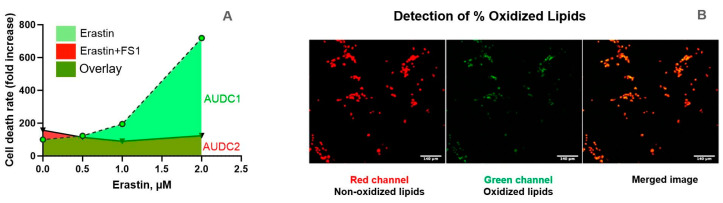
An example illustrating the calculation of the area under the dose curve (AUDC) in representative images. (**A**) AUDC is the integral area calculated under the curve of either LDH release or values LPO at all concentrations of ferroptosis inducers. Thus, this single parameter comprises the results obtained with all concentrations of either Erastin and or RSL3 representing an integrative parameter characterizing efficiency of both substances to induce ferroptosis. Such a parameter is particularly required if the cells strongly react to the small changes in inducer concentrations as it is the case in our study. (**B**) Determination of the levels of lipid peroxidation. The level of lipid peroxidation (% oxidized lipids) was determined from images as shown in B based on the determination of the fluorescens of oxidized (green) and non-oxidized (red) lipids in the BODIPY™ 581/591 C11 stained cells. The % oxidized lipids was calculated using the following equation: % oxidized lipids = intensity green/(intensity red + intensity green). Fluorescence analysis shows 1 representative set of images out of 4 independent experiments. Scale bar: 140 μm.

**Figure 2 antioxidants-14-00189-f002:**
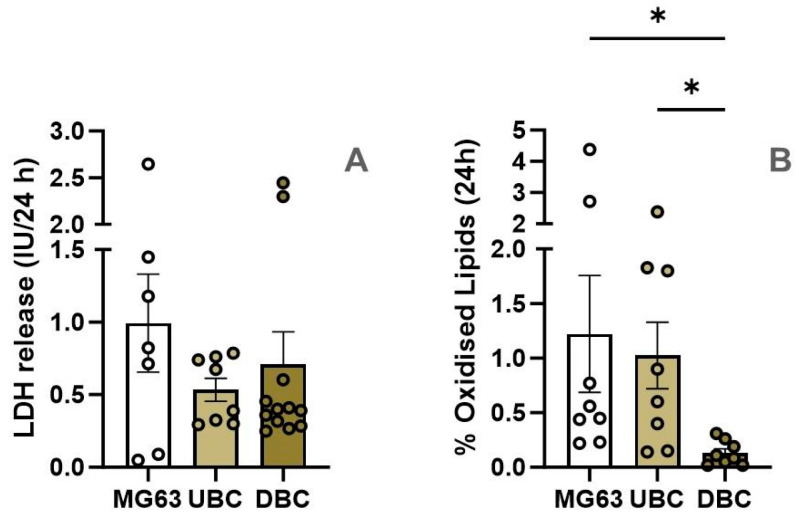
Baseline LDH release (**A**) and LPO levels (**B**) in three types of untreated cells. The cells were incubated in fresh incubation medium for 24 h. After 24 h the medium was collected to determine the LDH levels and cells were stained with BODIPY™ 581/591 C11 to determine lipid peroxidation. *—*p* ≤ 0.05. Statistical evaluation was performed with one way ANOVA followed by post hoc Holm–Sidak’s multiple comparisons test.

**Figure 3 antioxidants-14-00189-f003:**
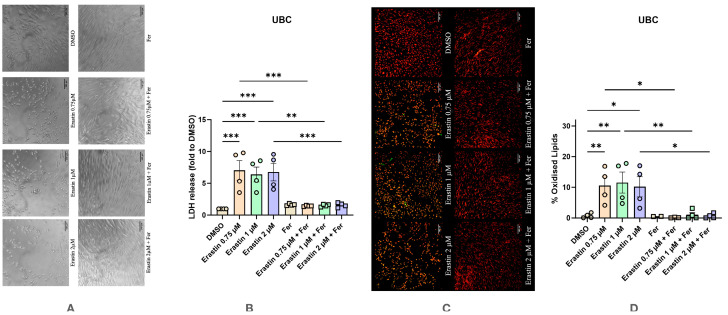
Effect of Erastin on undifferentiated human bone marrow mesenchymal stem cells with and without the addition of Ferrostatin-1. (**A**) Effect of Erastin on changes in cell morphology. The images were taken with a Zeiss LSM 510 microscope, 10× lens (scale bar: 140 μm). (**B**) Changes in levels of LDH release under various conditions. (**C**) Effect of Erastin on lipid peroxidation staining with BODIPY™ 581/591 C11. The images were taken with a Zeiss LSM 510 microscope, 10× lens using a red color filter for the non-oxidized form of the dye and green color filter for the oxidized form of the dye, after that two images were merged (scale bar: 140 μm). (**D**) Effect of Erastin on lipid peroxidation. Data are represented as means ± SEM (error bars), statistical significance was analyzed with RM-ANOVA followed by post hoc Holm–Sidak’s multiple comparisons test. n = 4. *—*p* ≤ 0.05; **—*p* ≤ 0.001; ***—*p* ≤ 0.0001.

**Figure 4 antioxidants-14-00189-f004:**
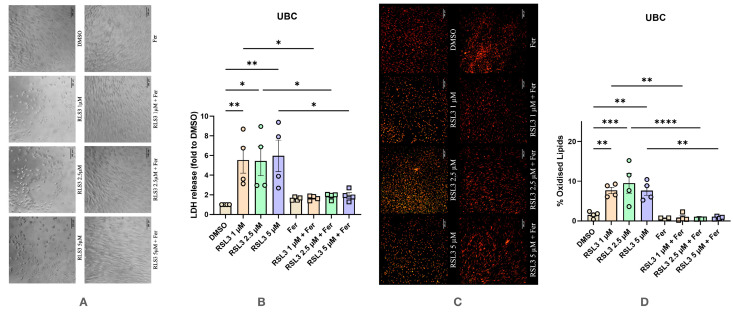
Effect of RSL3 on undifferentiated human bone marrow mesenchymal stem cells with and without the addition of Ferrostatin-1. (**A**) Effect of RSL3 on changes in cell morphology. The images were taken with a Zeiss LSM 510 microscope, 10× lens (scale bar: 140 μm). (**B**) Changes in levels of LDH release under various conditions. (**C**) Effect of RSL3 on lipid peroxidation staining with BODIPY™ 581/591 C11. The images were taken with a Zeiss LSM 510 microscope, 10× lens using a red color filter for the non-oxidized form of the dye and green color filter for the oxidized form of the dye, after the two images were merged (scale bar: 140 μm). (**D**) Effect of RSL3 on lipid peroxidation. Data are represented as means ± SEM (error bars), statistical significance was analyzed with RM-ANOVA followed by post hoc Holm–Sidak’s multiple comparisons test. n = 4. *—*p* ≤ 0.05; **—*p* ≤ 0.01, ***—*p* ≤ 0.001, ****—*p* ≤ 0.0001.

**Figure 5 antioxidants-14-00189-f005:**
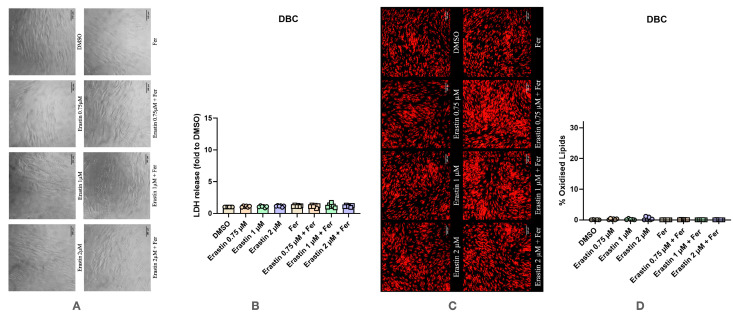
Effect of Erastin on differentiated human bone marrow mesenchymal stem cells (DBC) with and without the addition of Ferrostatin-1. (**A**) Effect of Erastin on changes in cell morphology. The images were taken with a Zeiss LSM 510 microscope, 10× lens (scale bar: 140 μm). (**B**) Changes in levels of LDH release under various conditions. (**C**) Effect of Erastin on lipid peroxidation staining with BODIPY™ 581/591 C11. The images were taken with a Zeiss LSM 510 microscope, 10× lens using a red color filter for the non-oxidized form of the dye and green color filter for the oxidized form of the dye, after that two images were merged (scale bar: 140 μm). (**D**) Effect of Erastin on lipid peroxidation. Data are represented as means ± SEM (error bars), statistical significance was analyzed with RM-ANOVA followed by post hoc Holm–Sidak’s multiple comparisons test. n = 6.

**Figure 6 antioxidants-14-00189-f006:**
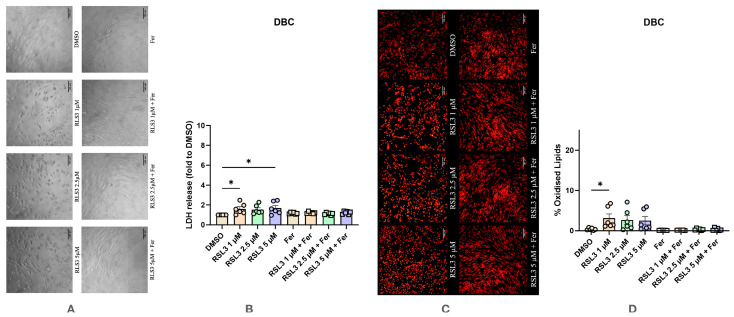
Effect of RSL3 on differentiated human bone marrow stromal cells (DBC) with and without the addition of Ferrostatin-1. (**A**) Effect of RSL3 on changes in cell morphology. The images were taken with a Zeiss LSM 510 microscope, 10× lens (scale bar: 140 μm). (**B**) Changes in levels of LDH release under various conditions. (**C**) Effect of RSL3 on lipid peroxidation staining with BODIPY™ 581/591 C11. The images were taken with a Zeiss LSM 510 microscope, 10× lens using a red color filter for the non-oxidized form of the dye and green color filter for the oxidized form of the dye, after that two images were merged (scale bar: 140 μm). (**D**) Effect of RSL3 on lipid peroxidation. Data are represented as means ± SEM (error bars), statistical significance was analyzed with RM-ANOVA followed by post hoc Holm–Sidak’s multiple comparisons test. n = 6. *—*p* ≤ 0.05.

**Figure 7 antioxidants-14-00189-f007:**
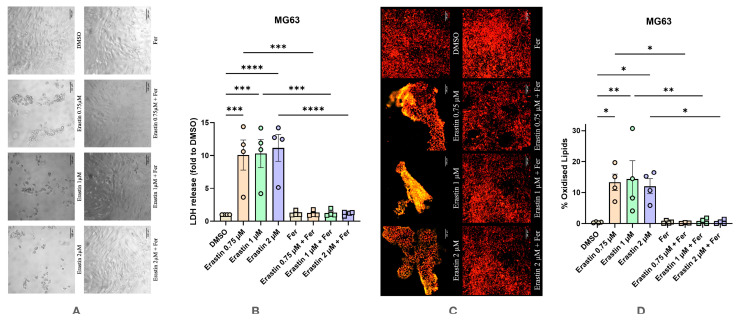
Effect of Erastin on human osteosarcoma cell line MG63 with and without the addition of Ferrostatin-1. (**A**) Effect of Erastin on changes in cell morphology. The images were taken with a Zeiss LSM 510 microscope, 10× lens (scale bar: 140 μm). (**B**) Changes in levels of LDH release under various conditions. (**C**) Effect of Erastin on lipid peroxidation staining with BODIPY™ 581/591 C11. The images were taken with a Zeiss LSM 510 microscope, 10× lens using a red color filter for the non-oxidized form of the dye and green color filter for the oxidized form of the dye, after that two images were merged (scale bar: 140 μm). (**D**) Effect of Erastin on lipid peroxidation. Data are represented as means ± SEM (error bars), statistical significance was analyzed with RM-ANOVA followed by post hoc Holm–Sidak’s multiple comparisons test. n = 4. *—*p* ≤ 0.05; **—*p* ≤ 0.01, ***—*p* ≤ 0.001, ****—*p* ≤ 0.0001.

**Figure 8 antioxidants-14-00189-f008:**
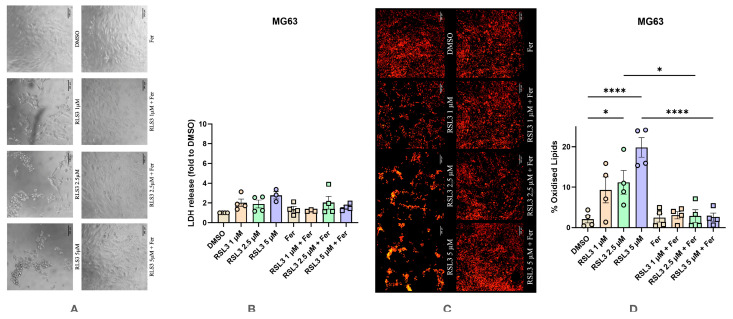
Effect of RSL3 on human osteosarcoma cell line MG63 with and without the addition of Ferrostatin-1. (**A**) Effect of RSL3 on changes in cell morphology. The images were taken with a Zeiss LSM 510 microscope, 10× lens (scale bar: 140 μm). (**B**) Changes in levels of LDH release under various conditions. (**C**) Effect of RSL3 on lipid peroxidation staining with BODIPY™ 581/591 C11. The images were taken with a Zeiss LSM 510 microscope, 10× lens using a red color filter for the non-oxidized form of the dye and green color filter for the oxidized form of the dye, after that two images were merged (scale bar: 140 μm). (**D**) Effect of RSL3 on lipid peroxidation. Data are represented as means ± SEM (error bars), statistical significance was analyzed with RM-ANOVA followed by post hoc Holm–Sidak’s multiple comparisons test. n = 4. *—*p* ≤ 0.05, ****—*p* ≤ 0.0001.

**Figure 9 antioxidants-14-00189-f009:**
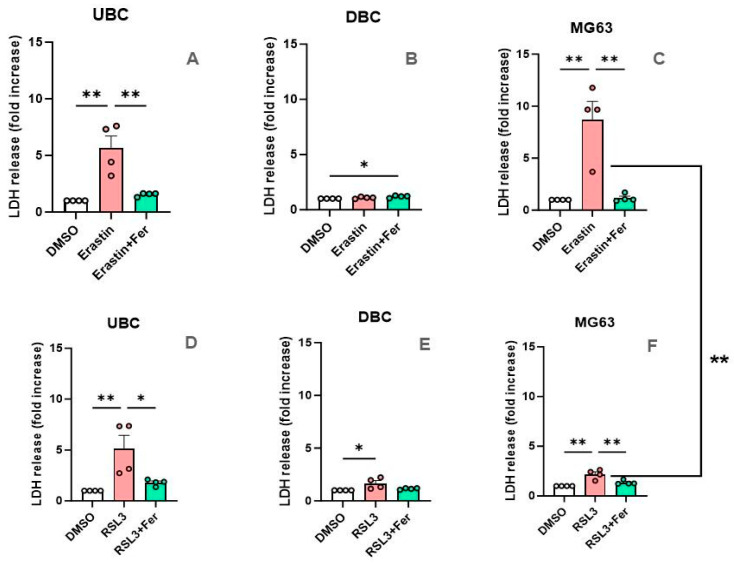
Area under dose-response curve for LDH analysis. (**A**) Effect of Erastin on UBC. (**B**) Effect of Erastin on DBC. (**C**) Effect of Erastin on MG63. (**D**) Effect of RSL3 on UBC. (**E**) Effect of RSL3 on DBC. (**F**) Effect of RSL3 on MG63. The data are presented as means ± SEM (error bars). Statistical evaluation was performed with one way ANOVA followed by post hoc Holm–Sidak’s multiple comparisons test. *—*p* ≤ 0.05; **—*p* ≤ 0.01.

**Figure 10 antioxidants-14-00189-f010:**
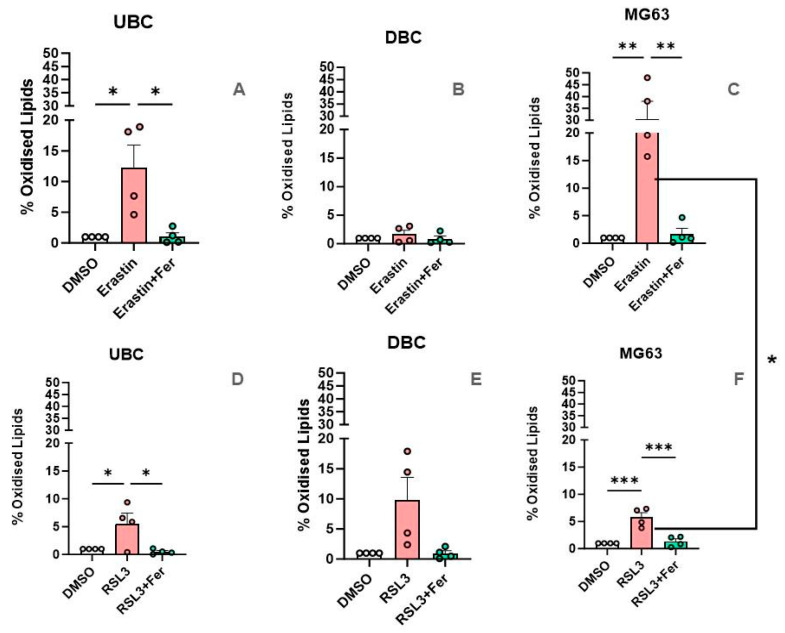
Area under dose-response curve for LPO analysis. (**A**) Effect of Erastin on UBC. (**B**) Effect of Erastin on DBC. (**C**) Effect of Erastin on MG63. (**D**) Effect of RSL3 on UBC. (**E**) Effect of RSL3 on DBC. (**F**) Effect of RSL3 on MG63 Data are presented as means ± SEM (error bars). Statistical evaluation was performed with one way ANOVA followed by post hoc Holm–Sidak’s multiple comparisons test. *—*p* ≤ 0.05; **—*p* ≤ 0.01; ***—*p* ≤ 0.001.

**Figure 11 antioxidants-14-00189-f011:**
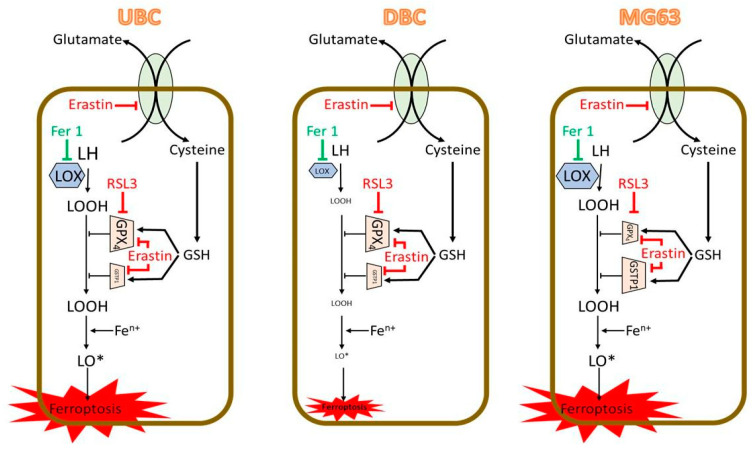
Mode of action of RSL3 and Erastin in tested bone cells. RSL3 inhibits specifically GPX4 and is expected to activate ferroptosis if the levels of LOOH are controlled predominantly by GPX4. Erastin inhibits glutamate-cysteine antiporter and is expected to activate ferroptosis if it is controlled either by GPX4 or by other GSH dependent enzymes, such as GSTP1. The fact that both RSL3 and Erastin induce ferroptosis in UBC suggests that it is controlled by GPX4. The fact that in MG63 ferroptosis is activated only by Erastin suggest that it is controlled by GSTP1. The elevated synthesis of LOOH is the obligatory prerequisite for induction of ferroptosis. If the initial levels of LOOH is low, as it was observed in DBC, then ferroptosis cannot be executed. This scheme illustrates the two major effector pathways.

## Data Availability

The data presented in this study are available on request from the corresponding author. The data are not publicly available because they should be handled only by persons with biomedical background.

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
