# Peer review of "Osteosarcoma Cells and Undifferentiated Human Mesenchymal Stromal Cells Are More Susceptible to Ferroptosis than Differentiated Human Mesenchymal Stromal Cells"

_antioxidants, 2025, doi:10.3390/antiox14020189_

Round 1
Reviewer 1 Report
The paper by Smirnova et al.,attempts to identify different ferroptosis mechanism in human undifferentiated bone marrow derived stromal cells, differentiated bone marrow stromal cells and a osteosarcoma cell line. By using different ferroptosis inducers (Erastin and RSL3) and performing rescue experiments with Ferrostatin-1 the authors provide preliminary evidences that GPX4 promotes ferroptosis in undifferentiated mesenchymal cells whereas GSTP1 rules the same process in osteosarcoma cells. On the contrary, differentiated mesenchymal cells do not undergo ferroptosis, but die by an undefined mechanism. Although interesting, this paper deserves imporvements as in detailed comments below.
Major
1. Lines 278 and following: the authors noted that ferroptosis is not implicated in DBC cell death. Experiments aimed to evaluate LPO-induced apoptosis and/or autophagy should be performed to identify DBC’s death mechanism
2. What about the expression level of GPX4 and GSTP1 in all the three cell lines?
3. Line 414: the authors assess that ferroptosis induction in osteosarcoma cells is probably due to GSTP1 activity. An experiment with siRNAs or CRISPR/Cas deltion of GSTP1 should be performed to validate this assumption.
Minor
1. Line 49 and following. The authors state that ferroptosis induction is a promising strategy to treat cancer and that it’s true also for osteosarcoma (ref. 5), indicating a series of drug inhibitors of ferroptosis as efficient against rhabdomyosarcoma [13,14], fibrosarcoma [15] and osteosarcoma [16,17]. . However, at line 56, they declare that ferroptosis induction is deleterious. May the authors mean ferroptosis “induction in normal cells”? If so, please specify because we have talked about cancer and the reader still feels to be in that field.
2. Line 80. Put a dot after ferroptosis and start another sentence with "Ferrostatin-1 is an amine."..etc
3. Figure1B, please indicate the number of IF experiments performed (imagese are representative of n independent experiments)
4. Lines 243-245: Please, Rephrase
5. In the discussion section, add some comments about the importance of the authors findings regarding tissue regeneration
Author Response
Response to the reviewer#1
Major comments
The paper by Smirnova et al.,attempts to identify different ferroptosis mechanism in human undifferentiated bone marrow derived stromal cells, differentiated bone marrow stromal cells and a osteosarcoma cell line. By using different ferroptosis inducers (Erastin and RSL3) and performing rescue experiments with Ferrostatin-1 the authors provide preliminary evidences that GPX4 promotes ferroptosis in undifferentiated mesenchymal cells whereas GSTP1 rules the same process in osteosarcoma cells. On the contrary, differentiated mesenchymal cells do not undergo ferroptosis, but die by an undefined mechanism. Although interesting, this paper deserves improvements as in detailed comments below.
Detail comments
Major
- Lines 278 and following: the authors noted that ferroptosis is not implicated in DBC cell death. Experiments aimed to evaluate LPO-induced apoptosis and/or autophagy should be performed to identify DBC’s death mechanism.
Thank you for this suggestion. Indeed, we did not determine the exact mechanism of cell death in DBC. We did not perform such experiments because the cell death rate in DBC was negligibly low compared to other cell types. However, we fully appreciate the reviewer’s concern and have now clearly stated this gap as a limitation of our study. We have added the following statement to the manuscript: "The cell death rate in DBC was very low and did not undergo the ferroptotic pathway. We did not identify the mechanism of cell death, which is a limitation of our study."
- What about the expression level of GPX4 and GSTP1 in all the three cell lines?
Thank you for bringing this to our attention. The expression of GPX4 has been confirmed in osteocytes at both the mRNA and protein levels (PMID: 37432277). It was also determined in MG63 cells (PMID: 38166505). Additionally, GSTP1 expression has been confirmed in osteocytes (PMID: 28469982) and osteosarcoma cells (PMID: 18701490). We have now referenced these studies in the introduction/discussion sections of our manuscript.
Line 79 to 81. It has been previously shown that osteocytes cultured in diabetic periodontitis exhibit suppressed GPX4 expression [29]. In MG63 osteosarcoma cells, decreased GPX4 ex-pression is also associated with the progression of ferroptosis [30].
Line 438-441. In several tumor tissues, >90% of active GSTs is due to GSTP1 activity, which is highly expressed in various types of cancer [47], including osteosarcoma cells [48], whereas in healthy tissues, among them bone cells, osteocytes [49], GSTP1 expression is usually low [33]
- Line 414: the authors assess that ferroptosis induction in osteosarcoma cells is probably due to GSTP1 activity. An experiment with siRNAs or CRISPR/Cas deletion of GSTP1 should be performed to validate this assumption.
Thank you for this important suggestion. To investigate the role of GSTP1, we conducted additional experiments using specific GSTP1 inhibitors. These inhibitors strongly induced cell death, which was only partially inhibited by ferrostatin, in contrast to classical ferroptosis inducers. We conclude that GSTP1 is involved in other vital cellular processes and may induce additional mechanisms of cell death. As mentioned earlier, we did not explore other mechanisms of cell death in this model, but we have clearly stated this as a limitation of our study. Please, see the corresponding Figures in the "Supplementary Materials" section.
Line 372-377. To better understand the mechanism of ferroptosis induction in MG63 cells we tested additionally one inhibitor of GPX4 (cisplatin) and two inhibitors of GSTP1 (Auranofin, and Piperlongumine) in this cell line. The data are displayed in supple-ment. We did not observe any induction of cell death by cisplatin (Fig. S1), while both GSTP1 inhibitors strongly elevated cell death (Fig. S2 and Fig. S3), which was only par-tially inhibited by ferrostatin, in contrast to classical ferroptosis inducers., RSL3 and Erastin.
Line 442-446. Data showing the effect of specific GSTP1 inhibitors on osteosarcoma cell death are presented in Supplementary Figure S2 and Supplementary Figure S3, showing that both inhibitors strongly activate cell death, which is only partially inhibited by ferro-statin, in contrast to classical ferroptosis inducers. We assume that GSTP1 may also induce other cell death mechanisms
Minor
1 Line 49 and following. The authors state that ferroptosis induction is a promising strategy to treat cancer and that it’s true also for osteosarcoma (ref. 5), indicating a series of drug inhibitors of ferroptosis as efficient against rhabdomyosarcoma [13,14], fibrosarcoma [15] and osteosarcoma [16,17]. . However, at line 56, they declare that ferroptosis induction is deleterious. May the authors mean ferroptosis “induction in normal cells”? If so, please specify because we have talked about cancer and the reader still feels to be in that field.
Thank you for pointing this out. Indeed, we intended to convey that ferroptosis induction in normal cells can be deleterious, whereas its induction in cancer cells can be beneficial. We have added a clarification to this sentence to the manuscript, which also addresses your comment #5.
Line 452-462. The induction of ferroptosis in bone cells can be either beneficial if induced in cancer cells or deleterious if induced in normal cells, as the latter can slow down re-generation and contribute to osteoporosis. Iron is a cofactor for many enzymes in-volved in collagen synthesis and extracellular matrix formation, both of which are es-sential for bone regeneration. However, elevated iron levels can suppress osteoblast activity, resulting in weaker bones and reduced regenerative capacity [51]. It has been shown that these negative effects are due to the activation of ferroptosis in bone cells. While it was previously unclear whether this is due to impaired functionality of dif-ferentiated osteoblasts or failure of undifferentiated cells to initiate regeneration, our data suggest that ferroptosis induction in undifferentiated bone cells, rather than dif-ferentiated osteoblasts, is the key factor involved in compromised tissue regeneration.
- Line 80. Put a dot after ferroptosis and start another sentence with "Ferrostatin-1 is an amine.". etc.
Thank you for your suggestion. We have corrected this issue.
Line 83. Ferrostatin-1 is an aromatic amine [31]
- Figure1B, please indicate the number of IF experiments performed (images are representative of n independent experiments)
Thank you for this observation. We have now indicated this is a representative images illustrating the way of calculation that we used to determine AUDC.
Line 192-193. An example illustrating the calculation of the Area Under the Dose Curve (AUDC) in representa-tive images.
- Lines 243-245: Please, Rephrase
Thank you for your helpful comment. Indeed, this section was not clearly written. We have revised it as follows:
Line 250-254. Morphological changes characteristic of cell death were not observed in the control group, but appeared upon treatment with all three concentrations of RSL3 (Fig. 4A), accompanied by an elevated release of LDH into the medium (Fig. 4B). These changes were reversed by the addition of Ferrostatin-1 (Fig. 4A, B). Additionally, LPO levels were elevated and returned to normal upon treatment with Ferrostatin-1 (Fig. 4C, D).
- In the discussion section, add some comments about the importance of the authors findings regarding tissue regeneration.
Thank you for your suggestion. We have added the following text to the discussion section along with your comment #1:
Line 452.462. The induction of ferroptosis in bone cells can be either beneficial if induced in cancer cells or deleterious if induced in normal cells, as the latter can slow down re-generation and contribute to osteoporosis. Iron is a cofactor for many enzymes involved in collagen synthesis and extracellular matrix formation, both of which are essential for bone regeneration. However, elevated iron levels can suppress osteoblast activity, resulting in weaker bones and reduced regenerative capacity [51]. It has been shown that these negative effects are due to the activation of ferroptosis in bone cells. While it was previously unclear whether this is due to impaired functionality of differentiated osteoblasts or failure of undifferentiated cells to initiate regeneration, our data suggest that ferroptosis induction in undifferentiated bone cells, rather than differentiated osteoblasts, is the key factor involved in compromised tissue regeneration.
Reviewer 2 Report
In this manuscript, the authors investigated whether ferroptosis differ between normal bone marrow stromal cells UBC, DBC and osteosarcoma cells MG63. Ferroptosis was induced by RSL3 or Erastin, while Ferrostatin-1 was used to inhibit ferroptosis. LDH and Lipid peroxidation were used to evaluate ferroptosis in cells. Their results suggested Erastin predominantly induced ferroptosis in MG63 cells, both RSL3 and Erastin induce ferroptosis in UBC, neither of them induced ferroptosis in DBC, indicating ferroptosis induction in UBC is primarily regulated by GPX4, while GSTP1 plays a key role in controlling ferroptosis in osteosarcoma cells. There are some concerns that need to be addressed.
All the cell morphology images are with low resolution and hard to tell the morphology difference between normal cells and ferroptosis cells. Please use higher magnification or resolution images with scale bar to present the cell morphology difference and use arrow to point out the cells with different morphology.
Erastin can induce ferroptosis of MG63 cells through GSTP1 signaling, however, if use another small molecule drug to treat MG63, e.g. cisplatin, the ferroptosis can be induced GPX4 signaling. Therefore, it is not appropriate to conclude that GSTP1 plays a key role in controlling ferroptosis in osteosarcoma cells. Please correct this conclusion.
Please provide other evidence such as western blot to demonstrate either GPX4 or GSTP1 signaling pathway in ferroptosis that induced by either RSL3 or Erastin.
It is not reasonable to compare LDH release and oxidized lipids in MG63 cells induced by Erastin and RSL3 in figure 9 and 10. The mechanisms of Erastin and RSL3 induced ferroptosis are different.
Author Response
Response to the Reviewer #2
Major comments
In this manuscript, the authors investigated whether ferroptosis differ between normal bone marrow stromal cells UBC, DBC and osteosarcoma cells MG63. Ferroptosis was induced by RSL3 or Erastin, while Ferrostatin-1 was used to inhibit ferroptosis. LDH and Lipid peroxidation were used to evaluate ferroptosis in cells. Their results suggested Erastin predominantly induced ferroptosis in MG63 cells, both RSL3 and Erastin induce ferroptosis in UBC, neither of them induced ferroptosis in DBC, indicating ferroptosis induction in UBC is primarily regulated by GPX4, while GSTP1 plays a key role in controlling ferroptosis in osteosarcoma cells. There are some concerns that need to be addressed.
All the cell morphology images are with low resolution and hard to tell the morphology difference between normal cells and ferroptosis cells. Please use higher magnification or resolution images with scale bar to present the cell morphology difference and use arrow to point out the cells with different morphology.
Thank you for this comment. We have improved the resolution of the images as much as it was possible and added the scale bars. Please, see the corresponding figures in the revised version of the manuscript.
Erastin can induce ferroptosis of MG63 cells through GSTP1 signaling, however, if use another small molecule drug to treat MG63, e.g. cisplatin, the ferroptosis can be induced GPX4 signaling. Therefore, it is not appropriate to conclude that GSTP1 plays a key role in controlling ferroptosis in osteosarcoma cells. Please correct this conclusion.
We thank reviewer for this excellent point. Indeed, the other inhibitors of GPX4 could do it. To address this issue we tested effect of cisplatin on the induction of cell death in MG63, but we did not observe any induction of cell death by this substance. Nevertheless, we take well the critic of the reviewer and formulated the conclusion in the following way. Please, see also the figures in the "Supplementary Materials" section.
These suggest that the induction of GPX4 mediated ferroptosis may operate in MG63, but it is only a weak mechanism compared to the induction via GSTP1.
Line 433-438. The effect of GPX4 inhibitor (cisplatin) on osteosarcoma cells is shown in Supplementary Figure S1. However, in this case, did not observe any induction of cell death by this substance, further supporting our conclusion that GSTP1 most likely plays an im-portant role in the induction of cell death in osteosarcoma. These suggest that the in-duction of GPX4 mediated ferroptosis may operate in MG63, but it is only a weak mechanism compared to the induction via GSTP1.
Please provide other evidence such as western blot to demonstrate either GPX4 or GSTP1 signaling pathway in ferroptosis that induced by either RSL3 or Erastin.
We thank reviewer for this important comment. To address this issue we performed additional experiments with specific GSTP1 inhibitors. We have shown that both inhibitors are strongly activated cell death, which was only partially inhibited by ferrostatin, in contrast to classical inducers of ferroptosis. We consider that GSTP1 is involved also in other vital for cell function processes and can induce also other mechanisms of cell death. As we mentioned above we did not study other mechanisms of cell death in this model, but we clearly indicate it as a limitation of our study. Please, see also the figures in the "Supplementary Materials" section.
Line 444-448. Data showing the effect of specific GSTP1 inhibitors on osteosarcoma cell death are presented in Supplementary Figure S2 and Supplementary Figure S3, showing that both inhibitors strongly activate cell death, which is only partially inhibited by ferro-statin, in contrast to classical ferroptosis inducers. We assume that GSTP1 may also induce other cell death mechanisms.
Detail comments
It is not reasonable to compare LDH release and oxidized lipids in MG63 cells induced by Erastin and RSL3 in figure 9 and 10. The mechanisms of Erastin and RSL3 induced ferroptosis are different.
Thank you for pointing this issue. We believe as well that they are different, but there still many reports considering that both substances induce ferroptosis via GRP1. The reason why we would like to keep these data presentation and we hope that we can convince the reviewer with this argument.

Round 2
Reviewer 1 Report
Dear Auhtors, this reviewer feels your paper improved. However, there are two further minor issues to address.
Figure 1 lacks of panel A and B and, as I previously suggested, you should add how many IF experiments were performed. Thus, at the end of Figure 1 legend, panel B, you should add: "Immunofluorescence analysis is representative of x experiments"
Figure 1 lacks of panel A and B and, as I previously suggested, you should add how many IF experiments were performed. Thus, at the end of Figure 1 legend, panel B, you should add: "Immunofluorescence analysis is representative of x experiments"
Author Response
Comment#1. Figure 1 lacks of panel A and panel B and, as I previously suggested , you should add how many IF experiments were performed. Thus, at the end of Figure 1 legend, Panel B, you should add: "Immunofluorescence analysis is representative of x experiments".
Answer of the authors. We thank the reviewer for this comment and apologize for not properly addressing this point. We have now introduced “Panel A” and “Panel B” in the figure legend and indicated the number of representative experiments as follows: “The fluorescence analysis shows 1 representative image set from 4 independent experiments.”
Reviewer 2 Report
The revised version addressed all of my concerns.
The revised version addressed all of my concerns.
Author Response
Comment#1. The revised version addressed all of my concerns
Answer of the authors. We thank the reviewer for approving our paper.